# A Preliminary Study of Deep Learning Sensor Fusion for Pedestrian Detection

**DOI:** 10.3390/s23084167

**Published:** 2023-04-21

**Authors:** Alfredo Chávez Plascencia, Pablo García-Gómez, Eduardo Bernal Perez, Gerard DeMas-Giménez, Josep R. Casas, Santiago Royo

**Affiliations:** 1Centre for Sensors, Instrumentation and Systems Development (CD6), Polytechnic University of Catalonia (UPC), Rambla de Sant Nebridi 10, 08222 Terrassa, Spain; 2Beamagine S.L. Carrer de Bellesguard 16, 08755 Castellbisbal, Spain; 3Image Processing Group, TSC Department, Polytechnic University of Catalonia (UPC), Carrer de Jordi Girona 1-3, 08034 Barcelona, Spain

**Keywords:** sensor fusion, convolutional neural networks: sensor calibration, autonomous driving

## Abstract

Most pedestrian detection methods focus on bounding boxes based on fusing RGB with lidar. These methods do not relate to how the human eye perceives objects in the real world. Furthermore, lidar and vision can have difficulty detecting pedestrians in scattered environments, and radar can be used to overcome this problem. Therefore, the motivation of this work is to explore, as a preliminary step, the feasibility of fusing lidar, radar, and RGB for pedestrian detection that potentially can be used for autonomous driving that uses a fully connected convolutional neural network architecture for multimodal sensors. The core of the network is based on SegNet, a pixel-wise semantic segmentation network. In this context, lidar and radar were incorporated by transforming them from 3D pointclouds into 2D gray images with 16-bit depths, and RGB images were incorporated with three channels. The proposed architecture uses a single SegNet for each sensor reading, and the outputs are then applied to a fully connected neural network to fuse the three modalities of sensors. Afterwards, an up-sampling network is applied to recover the fused data. Additionally, a custom dataset of 60 images was proposed for training the architecture, with an additional 10 for evaluation and 10 for testing, giving a total of 80 images. The experiment results show a training mean pixel accuracy of 99.7% and a training mean intersection over union of 99.5%. Also, the testing mean of the IoU was 94.4%, and the testing pixel accuracy was 96.2%. These metric results have successfully demonstrated the effectiveness of using semantic segmentation for pedestrian detection under the modalities of three sensors. Despite some overfitting in the model during experimentation, it performed well in detecting people in test mode. Therefore, it is worth emphasizing that the focus of this work is to show that this method is feasible to be used, as it works regardless of the size of the dataset. Also, a bigger dataset would be necessary to achieve a more appropiate training. This method gives the advantage of detecting pedestrians as the human eye does, thereby resulting in less ambiguity. Additionally, this work has also proposed an extrinsic calibration matrix method for sensor alignment between radar and lidar based on singular value decomposition.

## 1. Introduction

Deep learning (DL) has experienced rapid growth in recent years due to its universal learning approach, robustness, generalization, and scalability [1]. In the area of DL, recursive neural networks (RNN) and convolutional neural networks (CNN) are the most commonly used, with CNN being the most prevalent in many applications, including image classification, speech recognition, obstacle detection, computer vision, and sensor fusion, among others.

Moreover, autonomous driving cars (ADC) constitute a constantly progressing research field, and the direction towards fully automated public cars will likely become a reality soon. According to some official predictions, most cars are expected to be fully autonomous by the year 2035, [2]. In this scenario, one of the safety factors in ADC is the detection of pedestrians in different weather conditions and different circumstances. For instance, the proper recognition and detection of pedestrians in the vicinity of the ADC are crucial in order to avoid potential road accidents, [1,3]. It is believed that the correlation between ADC and the detection of pedestrians in the surroundings of autonomous vehicles is crucial in order to widen the use of this vehicle technology.

Detection using pure stereo vision systems in normal weather conditions is challenging due to intrinsic conditions such as intensity and light variations, as well as different pedestrian clothing. However, there has been a lot of work done using stereo vision systems for pedestrian detection. For example, a method using a stereo vision system for extracting, classifying, and tracking pedestrians is presented in [4]. This method is mainly based on four directional features with a classifier that increase robustness against the small affine deformations of objects.

In addition, a recent review of object detection for autonomous vehicles (AVs) was carried out in [5]. In this review, the capabilities of different sensors, such as radar, camera, ultrasonic, infrared, and lidar were analyzed in different weather situations. A suggestion for fusing different sensors for object detection in AVs was made, with the Kalman filter method being recommended.

Up to now, the review suggests that an individual sensor cannot handle different environmental situations such as rain, fog, snow, and so forth. This situation requires the fusion of different modalities of sensors to enhance the field of view of an individual sensor. Sensor fusion is a broad field whose applications range among military [6], medical [7], remote sensing [8], self-driving vehicles [9], mobile robots [10], and others. Sensor fusion can be classified according to the input type that is used in the fusion network, based on the source of the fused data, the sensor configuration, the fusion architecture, and both classical and deep-learning-based algorithms [11].

The detection of pedestrians using classical methods can be found in the literature. For instance, Shakeri et al. [12] use lidar and/or vision systems to enhance a region of interest (RoI) by fusing an RGB image with a depth stereo vision system for better pedestrian detection. Naoki et al. [13] use a 3D lidar pointcloud to create an information map of people in motion that surrounded the vehicle. Torresan et al. [14] fused thermal images with video to detect people in bounding boxes. Musleh et al. [15] used a stereo camera vision system and a laser scanner, as well as a sensor hybrid fusion method, to localize objects. Pedestrians were identified using poly-lines and pattern recognition related to leg positions, dense disparity maps, and UV disparity. The pedestrians were also identified in a bounding box.

Also, sensor fusion techniques involving CNNs for pedestrian detection have been explored in the literature. For example, in [16], a CNN that combined HHA (horizontal disparity, height above ground, and angle) from a lidar cloud with RGB images was used for detecting people. Additionally, a survey comparing state-of-the-art deep learning methods for pedestrian detection was presented in [17]. In this work, CNN methods such as the Region-Based CNN (R-CNN), the Fast R-CNN, the Faster R-CNN, the Single Shot Multi-Box Detector (SSD), and the You Only Look Once (YOLO) were evaluated and compared. However, the networks were trained using different datasets and tools, so the comparison may not be very accurate. Furthermore, the Faster R-CNN for multi-spectral pedestrian detection was used in [18], where thermal and color images were fused to provide the complementary information required for daytime and nighttime detection. A method based on the Faster Region-Based Convolutional Neural Network (R-CNN) for pedestrian detection at night time using a visible light camera was proposed by Jong et al. in [19]. In addition, an approach that combined a classical method such as a Histogram of Oriented Gradients (HOG), which is a feature descriptor, with deep learning to detect objects, showed better performance when compared to a single CNN. They also analyzed the extraction of features from a large dataset using a CNN; however, instead they used an HOG. The detection of objects in this work was done using bounding boxes (Gao, 2020).

So far, the methods mentioned previously have relied on placing bounding boxes around detected and classified objects. However, humans require the detection of pedestrians at the pixel level to better understand their surroundings, especially in the process of building autonomous cars, [20]. Thus, semantic segmentation refers to the action of classifying objects in an image at a pixel-level or pixel-wise. In other words, each pixel is classified individually and assigned to a class that best represents it. This gives a more detailed understanding of the imagery than image classification or object detection, which can be crucial in detecting people in autonomous driving cars and other fields such as robotics or image search engines [20,21].

Sensor fusion approaches based on semantic segmentation have been performed in [22]. In this approach, a semantic segmentation algorithm was used to effectively fuse 3D pointclouds with images. Another approach [21] used a network consisting of three CNN encoder–decoder sub-networks to fuse RGB images, lidar, and radar for road detection. Additionally, Khaled et al. in [23] use two networks, SqueezeSeg and PointSeg, for semantic segmentation and to apply a feature fusion level to fuse a 3D lidar with an RGB camera for pedestrian detection.

Semantic segmentation is a well-developed algorithm based on image data [23] and that leverages recent work on sensor fusion carried out in [21] to handle multi-modal fusion. In this work, an asymmetrical CNN architecture was used that consisted of three encoder sub-networks, where each sub-network was assigned to a particular sensor stream: camera, lidar, and radar. The stream data was then fused by a fully connected layer whose outputs were upsampled by a decoder layer that recovered the fused data by performing pixel-wise classification. The upsampling was done in order to recover the lost spatial information and generate a dense pixel-wise segmentation map that accurately captured the fine-grained details of the input image, [24]. The stream layers were designed to be as compact as possible, based on the complexity of the incoming data from the sensor. The convolutional neural network presented in [21] was used to detect roads in severe weather conditions. Therefore, an attempt to apply the same network for pedestrian detection was done in this work, but without success. Hence, the strategy had to be changed to use a similar methodology.

Thus, the main contribution of this paper is to explore the feasibility of fusing RGB images with lidar and radar pointclouds for pedestrian detection using a small dataset. It is widely recognized that larger datasets lead to better training, while small datasets can result in overfitting [25]. Nevertheless, studies [26,27] have shown that data size is not an obstacle to high-performing models. To address the previous issue, a novel and practical fully connected deep learning CNN architecture for semantic pixel-wise segmentation called SegNet [24] has been proposed. The network consists of three SegNet sub-networks that downsample the inputs of each sensor, a fully connected (fc) neural network (NN) that fuses the sensor data, and a decoder network that upsamples the data. Therefore, the proposed method focuses on detecting people at a pixel level. The task of identifying pedestrians is referred to as semantic segmentation and involves producing pixel-level classifications based on a dataset that has been labeled at the pixel level. Typically, there is only one class of interest, namely pedestrians. Moreover, the inclusion of radar in the fusion process gives the advantage of being able to detect pedestrians in severe weather conditions. Additionally, an extrinsic calibration method for radar with lidar and an RGB camera, based on the work done in [28,29], has been presented.

The remainder of this article is organized as follows. Section 2 deals with the sensor calibration between lidar and radar, where an extrinsic calibration matrix (T) was found based on the singular value decomposition (SVD). Section 3 shows the deep learning model architecture used in this work. Section 4 presents the results of the calibration method and the results after training the network. Finally, Section 5 summarizes the performance of the calibration method and the network, and conclusions are drawn, together with future work.

A GitHub ROS repository is available at [30].

## 2. Sensor Calibration

The proposed method of this work consisted of finding the extrinsic parameters between a multi-modal L3CAM lidar-RGB sensor from the company Beamagine [31] and a UMRR-96 Type 153 radar from the company Smartmicro [32]. Figure 1 illustrates the system. The L3CAM was already calibrated, so the focus of this method was to find the rotation matrix R from the radar frame $R_{f}$ to the lidar frame $L_{f}$ and the translation vector t from the origin $O_{R}$ of frame $R_{f}$ to the origin $O_{L}$ of frame $L_{f}$. The three reference frames, as well as the rotation and translation, can be seen in Figure 2.

### 2.1. Calibration Board

A calibration board, proposed in [28], was chosen. It consisted of three rectangular styrofoam pieces laid side by side on top of each other. The middle layer contained four circles that served as edge detectors for the lidar and camera sensors. Furthermore, a trihedral corner reflector was placed in the back of the calibration board at the intersection of the four circles. It is worth mentioning that the styrofoam did not affect the detection of the radar signal when it was reflected by the corner reflector.

The layout of the calibration board is illustrated in Figure 3, where two pieces of styrofoam were used, and the centers of the circles were used as four point descriptors whose intersection gave a point coordinate (x,y) of the location of the laser that matched the position of the trihedral corner reflector. The dashed black line shows the place of the corner reflector. Moreover, a single copper-plated trihedral corner reflector was made and is shown in Figure 4. The dimensions were chosen according to [33] and were such that the single areas of the corner reflector were larger compared to the radar wavelength.

Thus, the side length edge of the three isosceles triangles (a) was chosen to be 14 cm, and the base of the triangles $L=a\sqrt{2}$ was 19.7 cm. According to [33], the radar cross section (RCS), which is the measure of a target’s ability to reflect radar signals in the direction of the radar receiver, and it can be calculated with the following Equations (Equation 1) and (Equation 2).
(1)$$A_{eff}=\frac{a^{2}}{\sqrt{3}}$$
(2)$$\sigma =\frac{4\pi a^{2}}{3\lambda^{2}}$$
where a is the length of the side edges of the three isosceles triangles, λ is the wavelength of the radar, A${}_{eff}$ is the effective area, and σ is the radar cross section.

In accordance with the datasheet of the UMRR-96 Type 153, the radar frequency was between 77–81 GHz; taking the average of these frequencies at 79 GHz yielded a λ = 3.79484 mm width, an A${}_{eff}$ = 0.011316 m${}^{2}$, and a σ = 111.74 m${}^{2}$.

### 2.2. Location and Calibration

In order to find the extrinsic parameters between lidar and radar, it is necessary to set two reference points: one for the lidar and one for the radar. To this end, the reference point location of the lidar is the crossing point center of the four circles, and the corner reflector is the reference point of the radar.

Since the L3CAM had already been calibrated, the rotation matrix and translation vector for the camera to lidar were available. Three procedures were followed to obtain lidar–radar target calibration points. First, the middle point of the intersection of the circles was calculated in the RGB image. Second, this point was transferred into the lidar frame. Third, the extrinsic parameters from the radar to the lidar frames, such as R and t, were calculated. The radar driver provided the location of the radar point.

#### 2.2.1. Step 1: Image Point Segmentation

Figure 5 shows the calibration board and the corner reflector, as well as the lidar and radar frames. Figure 6 depicts the four circles’ centers ($x_{i},y_{i}$), the middle point (x, y), and the two lines ($l_{1},l_{2}$). Based on classical analytical geometry [34], two equations that correspond to $l_{1},l_{2}$ were obtained—Equations (Equation 3) and (Equation 4). Then, by solving the previous equations, a crossing point (x, y) in pixels was found, as shown by Equations (Equation 5) and (Equation 6).
(3)$$\left(y_{2}-y1\right)x-\left(x_{2}-x1\right)y+\left[y_{1}\left(x_{2}-x_{1}\right)-x_{1}\left(y_{2}-y_{1}\right)\right]=0$$
(4)$$\left(y_{4}-y3\right)x-\left(x_{4}-x3\right)y+\left[y_{3}\left(x_{4}-x_{3}\right)-x_{3}\left(y_{4}-y_{3}\right)\right]=0$$
(5)$$x=\frac{-C_{1}B_{2}+C_{2}B_{1}}{A_{1}B_{2}-A_{2}B_{1}}$$
(6)$$y=\frac{-A_{1}C_{2}+A_{2}C_{1}}{A_{1}B_{2}-A_{2}B_{1}}$$
where $A_{1}=y_{2}-y_{1}$, $B_{1}=x_{2}-x_{1}$, $C_{1}=y_{1}\left(x_{2}-x_{1}\right)-x_{1}\left(y_{2}-y_{1}\right)$, $A_{2}=y_{4}-y_{3}$, $B_{2}=x_{4}-x_{3}$, and $C_{2}=y_{3}\left(x_{4}-x_{3}\right)-x_{3}\left(y_{4}-y_{3}\right)$.

#### 2.2.2. Step 2: Lidar Point

The first Algorithm 1 shows the process of how the point center (x,y) in the RGB image is transformed into the lidar frame, e.g. ${\left[x_{l},y_{l},z_{l}\right]}^{T}$. First, the OpenCV undistortPoints function is used to remove distortion from a set of image points (x,y). The function takes distorted points, camera matrix, and distortion coefficients as the inputs and outputs undistorted points $\left(u_{u},v_{u}\right)$. Then, the directional vector ${\left[u_{d},v_{d},1\right]}^{T}$ from the camera frame to the middle point (x,y) is computed by multiplying the undistorted point with the inverse intrinsic camera matrix. Next, the undistorted point is rotated to the lidar frame by multiplying it with ${}_{c}^{l}R$, which is the rotational matrix from the camera to the lidar frames. Now, the vector ${\left[u_{l},v_{l},1\right]}^{T}$ points to the center point (x,y). The next step is to calculate the crossing point of ${\left[u_{l},v_{l},1\right]}^{T}$ with the lidar plane that corresponds to the center of the board. The function PlaneLineCrossingPoint($u_{l},v_{l},PC_{l}$) is responsible for calculating the crossing point. It receives the lidar pointcloud $PC_{l}$ and the point $\left(u_{l},v_{l}\right)$ and outputs the 3D crossing point $\left(x_{l},y_{l},z_{l}\right)$.
**Algorithm 1** Lidar point coordinates $x_{l},y_{l},z_{l}$1:**procedure**2:    •Get the u,v undirstorted point in pixel coordinates.3:    **function** UndirstortPoint(x,y)4:        $u_{u}\leftarrow x$5:        $v_{u}\leftarrow y$6:        **return** $u_{u},v_{u}$7:    **end function**8:    •Get the directional vector in the camera frame.9:    **function** DirectionalVector($u_{u},v_{u}$)10:        $u_{d}\leftarrow u_{u}$11:        $v_{d}\leftarrow v_{u}$12:        $w_{d}\leftarrow 1$13:        **return** $u_{d},v_{d},w_{d}$14:    **end function**15:    •Get the directional vector in the lidar frame.16:    **function** Camera2LidaRotation($u_{d},v_{d},w_{d}$)17:        $\left[\begin{array}{c} u_{l}\\ v_{l}\\ w_{l} \end{array}\right]\leftarrow \begin{array}{c} {}_{c}^{l}R \end{array}{\left[\begin{array}{ccc} u_{d} & v_{d} & w_{d} \end{array}\right]}^{T}$18:        **return** $u_{l},v_{l},w_{l}$19:    **end function**20:    •Compute the intersection point between a line and a plane.21:    **function** PlaneLineCrossingPoint($u_{l},v_{l},w_{l}PC_{l}$)22:        $\vec{u}\leftarrow \left[u_{l},v_{l},w_{l}\right]$▷ creates the $\vec{u}$ vector23:        $Ax+By+Cz+D=0\leftarrow PC_{l}$▷ board’s parallel plane model24:        $\vec{n}\leftarrow \left[A,B,C\right]$▷ plane normal vector25:        $v_{o}\leftarrow \left[A,B,C,D\right]$▷ a point in the model plane26:        $p_{o}\leftarrow {\;}_{c}^{l}t$▷ translation vector from camera to lidar27:        $w\leftarrow p_{o}-v_{o}$28:        $D\leftarrow \vec{n}\cdot \vec{u}$▷ dot product29:        $N\leftarrow -\vec{n}\cdot \vec{w}$▷ dot product30:        $sl\leftarrow \frac{N}{D}$31:        $\vec{p}\leftarrow p_{o}+sl\times \vec{u}$▷ intersection point32:        **return** $\vec{p}$33:    **end function**34:**end procedure**

#### 2.2.3. Step 3: Extrinsic Calibration Matrix

The extrinsic calibration matrix T is composed of a rotation matrix R and a translation vector t that best aligns two data sets of points, as shown in Equation (Equation 7). The proposed alignment method for 3D data points is based on [29,35].

The alignment problem formulation can be stated as follows: Given two set of points $Y=\{y_{1},\cdots ,y_{n}\}$ and $X=\{x_{1},\cdots ,x_{n}\}$ with known correspondences, the task is to find a rigid transformation that optimally minimizes the sum of squared errors as seen in Equation (Equation 8).
(7)$$\begin{array}{c} T \end{array}=\left[\begin{array}{cc} R & t\\ 0 & 1 \end{array}\right]$$
(8)$$min\leftarrow \sum_{i=1}^{n}{∥y_{i}-Rx_{i}-t∥}^{2}$$

Algorithm 2 shows the procedure for obtaining T using singular value decomposition (SVD). Thus, the function AlignmentSVD(X,Y) takes as inputs the two data point sets *X* and *Y*. Then, the centroids or means $x_{0}$ and $y_{0}$ are calculated. Next, the cross-covariance matrix of the mean-reduced coordinates is calculated, and SVD is applied to *H*. Finally, the function returns the extrinsic parameter matrix T.
**Algorithm 2** Least squares alignment of two 3D point sets based on SVD1:**procedure**2:    Let $Y=\{y_{1},\cdots ,y_{n}\}$ and $X=\{x_{1},\cdots ,x_{n}\}$ two sets with corresponding points.3:    **function** AlignmentSVD(X,Y)4:        $y_{0}=\frac{1}{n}\sum_{i=1}^{n}y_{i}$▷ mean of the data point set *X*5:        $x_{0}=\frac{1}{n}\sum_{i=1}^{n}x_{i}$▷ mean of the data point set *Y*6:        $H=\sum_{i=1}^{n}\left(y_{i}-y_{0}\right){\left(x_{i}-y_{0}\right)}^{T}$▷ cross covariance between *X* and *Y*7:        $UDV^{T}=SVD\left(H\right)$▷ singular value decomposition of *H*8:        $R=VU^{T}$▷ rotation matrix9:        $t=y_{0}-Rx_{0}$▷ translation vector10:        T←[R,t]▷ extrinsic parameter matrix11:        **return** T12:    **end function**13:**end procedure**

Moreover, the matrix T can be used to transform the two pointclouds into the same RF. For instance having *Y* in a certain RF, *X* can be brought to *Y*’s RF by Equation (Equation 9).
(9)$$\tilde{X}=TX$$

## 3. Proposed Architecture

The proposed CNN network to handle the sensor fusion approach is based on [21,24]. Figure 7 illustrates the SegNet architecture used in this work. It consists of an encoder and a decoder network, with 26 convolutional, 5 pooling, and 5 upsampling layers, respectively. The function of the encoder network (EN) is to downsample the inputs and extract features. On the other hand, the decoder network (DN) upsamples the data and reconstructs the image. A fully connected layer between the EN and the DN is discarded with the purpose of retaining higher resolution feature maps at the deepest encoder output.

Inspired by the work in [21], a convolutional network was proposed to carry out a multi-modal sensor fusion approach to detect people. Figure 8 presents a convolutional neural network that consists of three ENs as inputs, a fully connected neural network (FCNN) that fuses the data, and a DN that outsamples and recovers the fused data.

Table 1 shows in detail the model architecture for the encoder, whereas Table 2 illustrates the input and output of the fully connected layer. Table 3 also shows the details of the decoder network model.

## 4. Results

The system used to handle the simulations was composed of a L3CAM lidar, a UMRR-96 Type 153 radar, and a GE66 Raider Intel ®Core(TM) i9-10980HK CPU with an NVIDIA GeForce RTX 3070 8Gb GPU. The Robot Operating System (ROS1) Noetic on Ubuntu 20.04.5 LTS was used to collect sensor data, compute extrinsic parameters, and align the sensors. Moreover, the CNN network was simulated in a Jupyter Notebook using Python 3 and a conda environment. More specifically, the simulations were divided into two parts: an extrinsic parameters matrix and network simulations.

### 4.1. Extrinsic Parameters Matrix

A ROS1 wrapper was developed to collect lidar data, and the ROS1 SmartMicro driver was used to gather radar data. As a result, a dataset of 11 lidar–RGB–radar (LRR) images was taken, 8 of which were taken indoors and 3 of which were taken outdoors. Figure 9 shows the RGB image of the calibration board, and Figure 10 illustrates in white the lidar pointcloud and in colored cubes, as well as the more sparse radar pointcloud. It is worth mentioning that the radar was used in long-range mode to reduce noise and better detect the corner detector, as is shown by the purple cube in Figure 10, placed almost in the middle of the four circles.

Then, Algorithm 1 was applied to each lidar–RGB image to obtain the lidar point coordinates $\left(x_{l},y_{l},z_{l}\right)$ of the middle point of the four circles. On the other hand, the ROS1 radar driver gives the coordinates of the corner detector placed behind the middle point. Figure 11 depicts the position of the lidar point as a blue sphere, while the position of the radar’s coordinate is shown as a brown cube. The colored pointcloud is the Ax+By+Cz+D=0 plane parallel to the board.

The two-point datasets, for example lidar and radar point sets, are shown in Figure 12. It can be seen that the two dataset of points were not aligned. Therefore, alignment was necessary. In other words, Algorithm 2 was applied to the two datasets to find the extrinsic matrix parameters T. Afterwards, Equation (Equation 9) was applied to the radar dataset to align and transfer it to the lidar frame $L_{f}$. Figure 13 shows the radar set after correction.

The norm was calculated in both lidar and radar datasets before and after correction by Equation (Equation 9) as shown in Figure 14. It shows the first eight images taken inside a laboratory; therefore, their distances fluctuated between 1.5 m and 2.0 m. The last three images were taken outside in a parking lot, where distances fluctuated around 14.0 m. Moreover, it can be seen by the blue line that the norms were reduced. Also, the figure shows the mean values of the two norms. As expected, the mean of the norm after correction was smaller than the mean of the norm before correction.

The transformation matrix T was applied to the radar coordinates given by the corner reflector, as can be seen by the blue sphere in Figure 11. Moreover, an outdoor LRR image was taken from a parking lot with people, as illustrated in Figure 15. The colored spheres represent the radar dataset before correction, and the white spheres represent the radar dataset after correction by Equation (Equation 9).

### 4.2. Network

The CNN was trained using a custom dataset that was taken at a parking lot with people walking in front of the system. The dataset was taken in the format yyyy-MM-dd:hh:mm:ss:zz, where yyyy denotes the year, MM the month, dd the day, hh the hour, mm the minute, ss the second, and zz the milliseconds. This format was chosen because the lidar, RGB, and radar frequencies are 6 Hz, 10 Hz, and 18 Hz, respectively. For instance, a typical format is 20221017_131347_782.pcd where hh = 13, mm = 13, and ss = 48. This means 48 will be repeated 6 times for the lidar, 10 times for the RGB, and 18 times for the radar. The value of zz = 782 will be different for each reading, making it easier to synchronize them.

Afterward, the dataset was synchronized, producing a total of 224 LRR images. Then, 30 LRR images were selected and expanded to 60 by flipping and adding them to the original set. The program ‘labelme’ was used to label the RGB images as a ‘single-class person’. In other words, ‘labelme’ can create polygons of the persons and save them as a JSON, which can be extracted as a PNG image with a single person class. The LiDAR images were saved in PCD format, extracted, and converted to 2D grayscale images with a 16-bit depth. They were interpolated to improve their quality. The radar images were also saved in PCD format, extracted, and converted to 2D grayscale images with a 16-bit depth. The location of each point in the radar images was projected vertically.

The training, evaluating, and testing modes were executed on Jupyter Notebook with Pytorch version 1.11 and the GPU activated. The LRR images were downsampled to a size of 256 × 256. Moreover, due to the capacity of the GPU, which is 8 GB, a batch size was set to one. Next, the network was trained for 300 epochs. Additionally, 10 LRR images were used for validation and 10 LRR for testing, giving a total dataset of 80 LRR images.

The results of the training mode are shown from Figure 16, Figure 17, Figure 18, Figure 19 and Figure 20. Figure 16 shows the RGB image that has been applied to the EN, whereas Figure 17 illustrates the ground truth image. Moreover, the lidar and radar images are shown in Figure 18 and Figure 19. Finally, Figure 20 depicts the resulting fused image between the lidar, radar, and RGB. Furthermore, the time elapsed during the training of the network was 7019.40 s, while the mean average of the model in testing mode was 0.010 s.

The target and prediction during training were stored in a vector. Later on, the intersection over union (IoU) was evaluated for each frame to see the percentage of the overlapping area between the prediction and the target. To this end, to evaluate the results of the network in training mode, pixel accuracy and intersection over union (IoU) metrics for the image segmentation were used. Additionally, the cross-entropy loss function was applied.

The IoU refers to the ratio of the overlap area in pixels to the union of the target mask and the prediction mask in pixels and is represented by Equation (Equation 10).The pixel accuracy refers to the ratio of the correctly identified positives and negatives to the size of the image and is represented by Equation (Equation 11).

Both Equations (Equation 10) and (Equation 11) are defined in terms of pixel accuracy.
(10)$$\begin{array}{cc} IoU & =\frac{TP}{TP+FP+FN};\\ \; & \end{array}$$
(11)$$\begin{array}{cc} Pixel\;Accuracy & =\frac{TP+TN}{TP+TN+FP+FN}, \end{array}$$
where the true positive (TP) is the area of intersection between the ground truth (GT) and the segmentation mask (S). The false positive (FP) is the predicted area outside the ground truth. The false negative (FN) is the number of pixels in the ground truth area that the model failed to predict. These parameters can be obtained using Equation (Equation 12). In this equation, if the confusion vector is equal to 1, this means that there is an overlapping; then the TP is computed by adding all the ones. The same procedure happens for the FP, TN, and FN.
(12)$$\begin{array}{ccc} confusion\;vector & = & \frac{prediction\;image}{label\;image}\\ TP & = & \sum if(confusion\;vector=1)\\ FP & = & \sum if(confusion\;vector=inf)\\ TN & = & \sum if(confusion\;vector=isnan)\\ FN & = & \sum if(confusion\;vector=0) \end{array}$$

Figure 21 shows the IoU versus pixel accuracy for over 60 LRR fused images. The average loss over the training of 300 epochs is shown in Figure 22.

The results of the evaluation mode are shown in Figure 23, where the blue line indicates the average loss of the training mode and the orange line indicates the average evaluation loss. It can be seen that the model started overfitting around epoch 50, with a lowest loss value of 1.411. At this point, the model was saved for use in the testing mode. In addition, the IoU and the pixel accuracy for the testing mode is shown in Figure 24, where the mean of the IoU was 94.4%, and the pixel accuracy was 96.2%.

The results of the testing mode are displayed in Figure 25, where the ground truth and the model’s output are overlaid. The model’s detection of pedestrians is shown in red, which represents the overlapping area with the ground truth. The white spots near and between the pedestrians are part of the model’s output but do not overlap with the ground truth. Light blue depicts weakly detected pedestrians, which was a result of the model’s overfitting and the limited size of the training data. This can be confirmed by the loss of 1.413 between the output and ground truth.

## 5. Conclusions

In conclusion, this article explored, as a preliminary or pilot step, the feasibility of fusing lidar, radar pointclouds, and RGB images for pedestrian detection by using a sensor fusion pixel semantic segmentation CNN SegNet network. The proposed method has the advantage of detecting pedestrians at the pixel level, which is not possible with conventional methods that use bounding boxes.

Experimental results using a custom dataset showed that the proposed architecture achieved high accuracy and performance, making it suitable for systems with limited computational resources. However, since this is a preliminary study, the results must be interpreted with caution.

An extrinsic calibration method based on SVD for lidar and radar in a multimodal sensor fusion architecture that includes an RGB camera was also proposed and was demonstrated to improve the mean norm between the lidar and radar points by 38.354%.

For future work, we look to investigate the behavior of the proposed architecture with a larger dataset and to test the model in different weather conditions. Additionally, evaluating the architecture under a real scenario by mounting the sensors on a driving car shall be investigated. Overall, this preliminary study provides a promising starting point for further research and development in the field of sensor fusion for pedestrian detection.

## Figures and Tables

**Figure 1 sensors-23-04167-f001:**
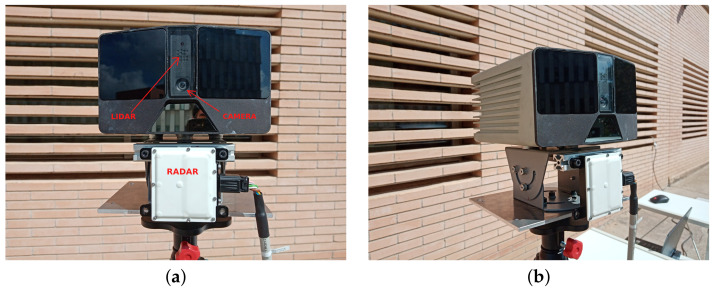
The sensors used on the testbench include: (**a**) a L3CAM sensor consisting of a lidar and an RGB camera on the top and (**b**) a UMRR-96 Type 153 radar sensor at the bottom.

**Figure 2 sensors-23-04167-f002:**
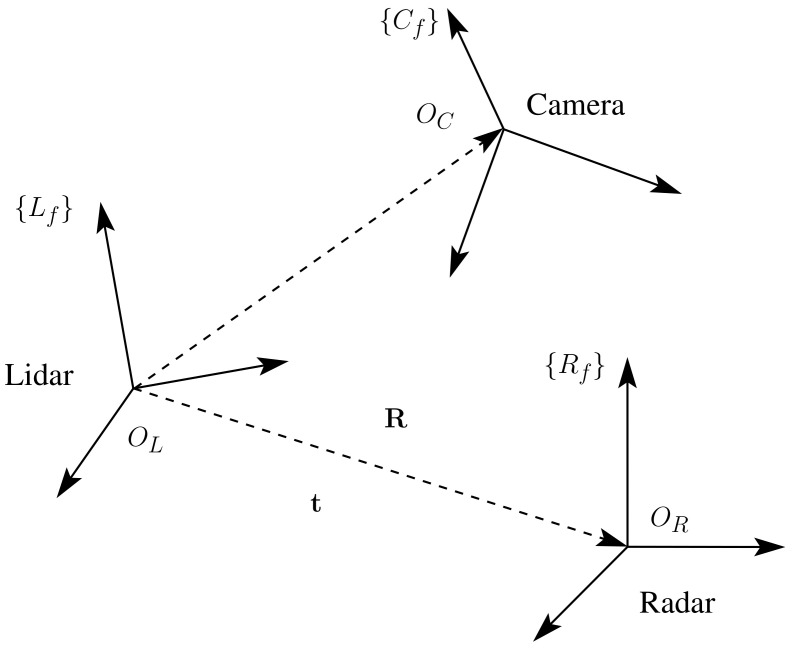
A schematic overview of the three frames: lidar, radar, and camera. A rotation matrix R and translation vector t from the radar to the lidar frames are also shown.

**Figure 3 sensors-23-04167-f003:**
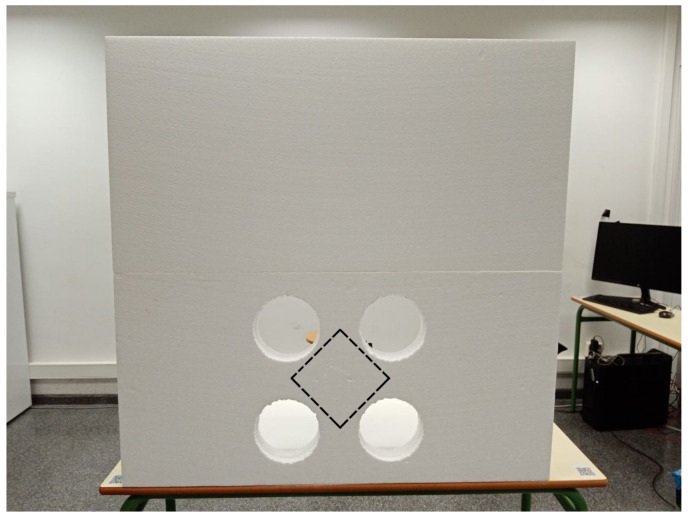
The styrofoam calibration board has black dashed lines that indicate the location of the corner reflector, which was placed in the center of the back of the board.

**Figure 4 sensors-23-04167-f004:**
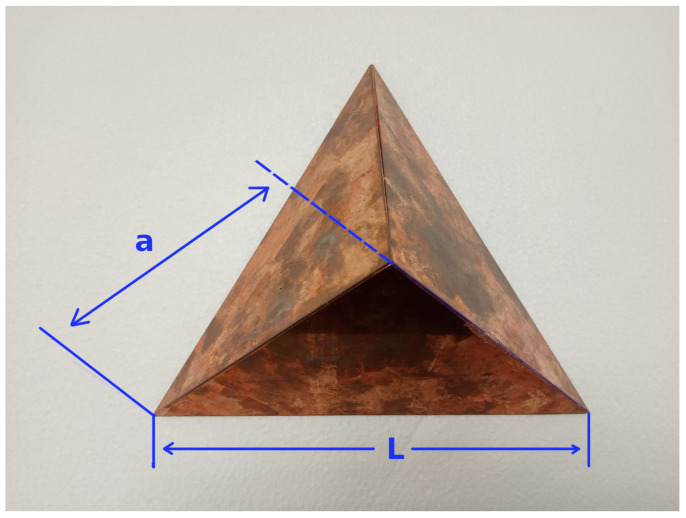
A custom trihedral corner reflector made of copper plates showing the side length edge of the three isosceles triangles (a) and the base of the triangles (L).

**Figure 5 sensors-23-04167-f005:**
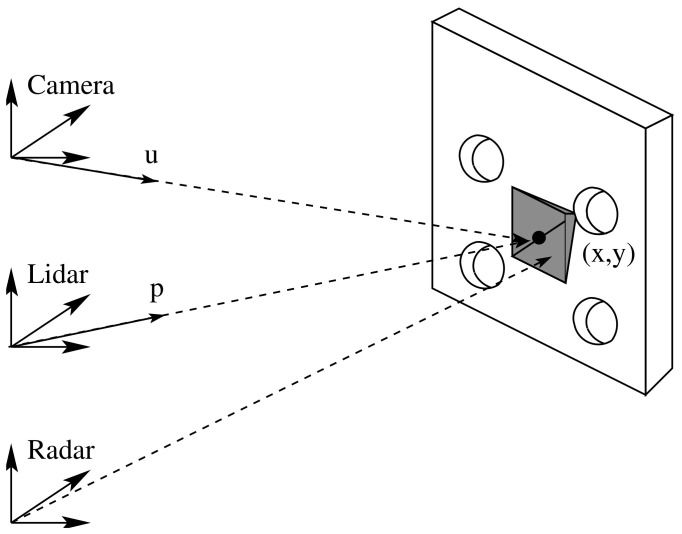
The figure shows the lidar and radar frames detecting the board center and the corner reflector, respectively.

**Figure 6 sensors-23-04167-f006:**
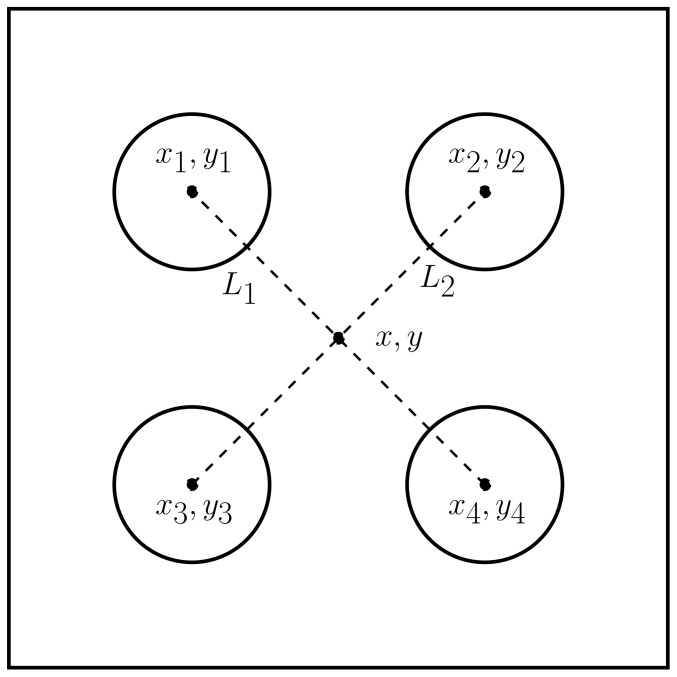
The figure shows the centers of four circles, two lines ($l_{1},l_{2}$), and the middle point (x,y).

**Figure 7 sensors-23-04167-f007:**
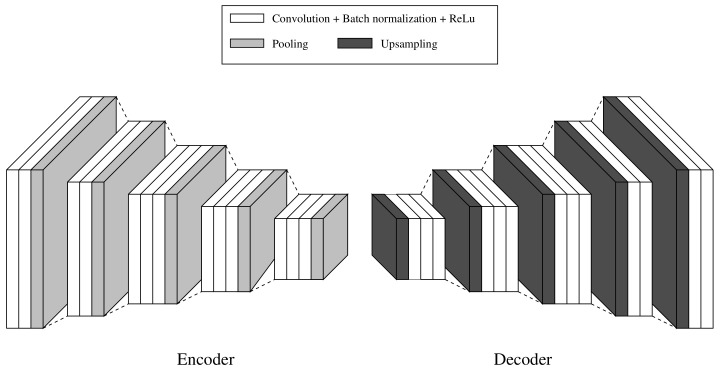
The pixel semantic segmentation SegNet CNN network used in the article. The encoder is placed on the left, while the decoder is placed on the right.

**Figure 8 sensors-23-04167-f008:**
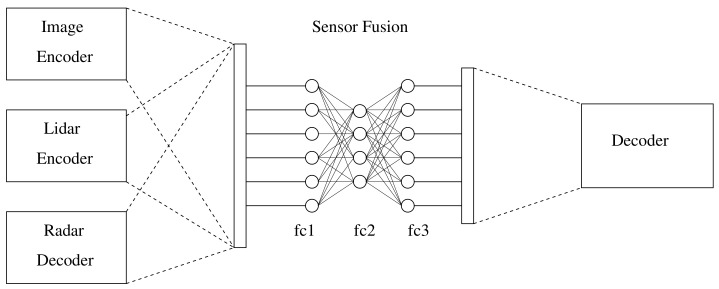
The architecture model consists of three decoder sub-networks, one for each sensor, a fully connected neural network, and a decoder.

**Figure 9 sensors-23-04167-f009:**
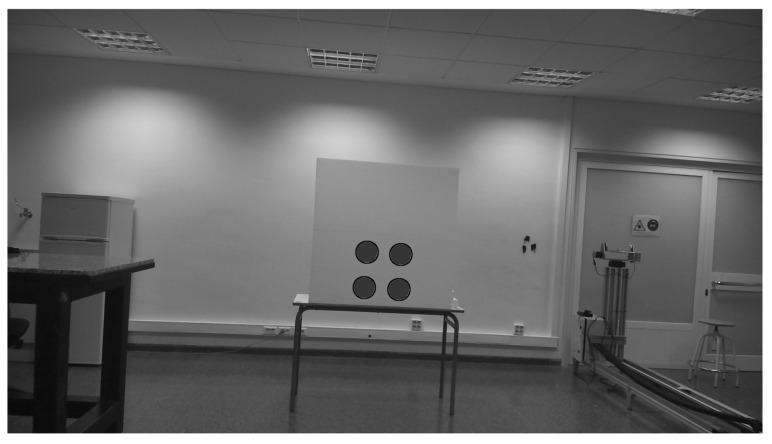
The RGB image of the calibration board.

**Figure 10 sensors-23-04167-f010:**
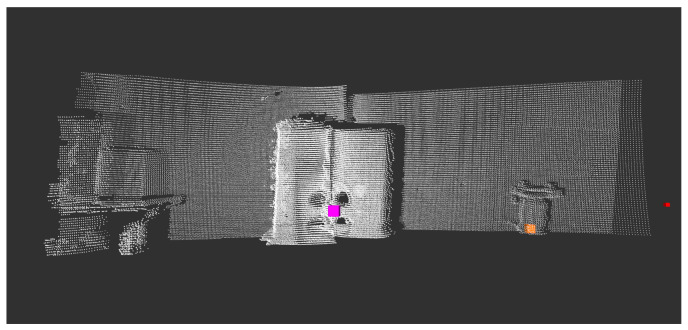
The lidar pointcloud is white, whereas the radar pointcloud, which is more sparse, is shown in colored cubes.

**Figure 11 sensors-23-04167-f011:**
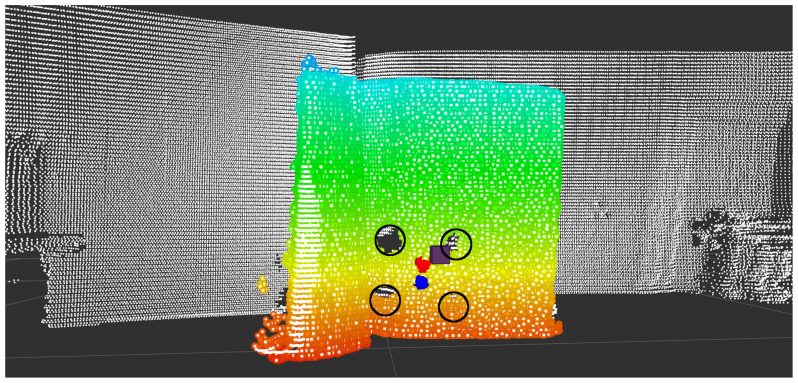
The blue sphere represents the lidar center position point; the brown cube represents the radar center position point; and the red sphere represents the aligned radar point. The colored pointcloud is the board’s parallel plane model.

**Figure 12 sensors-23-04167-f012:**
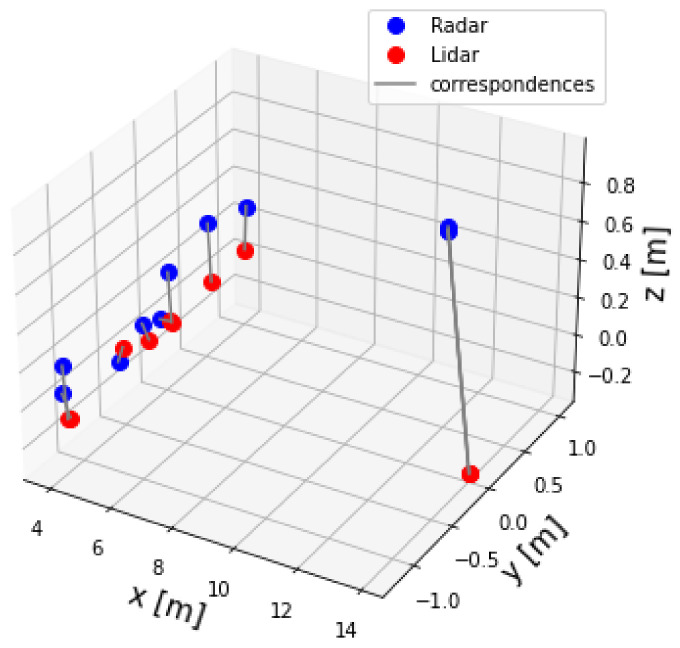
Lidar and radar pointsets before correction.

**Figure 13 sensors-23-04167-f013:**
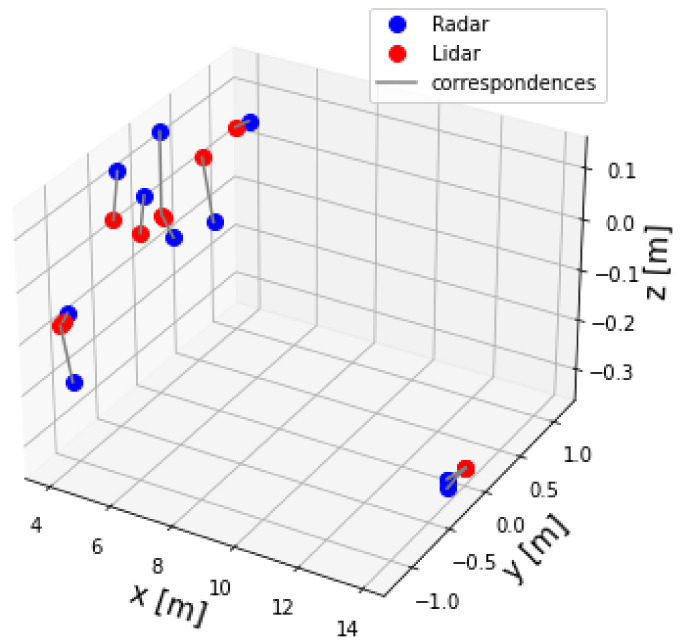
Lidar and radar pointsets after correction.

**Figure 14 sensors-23-04167-f014:**
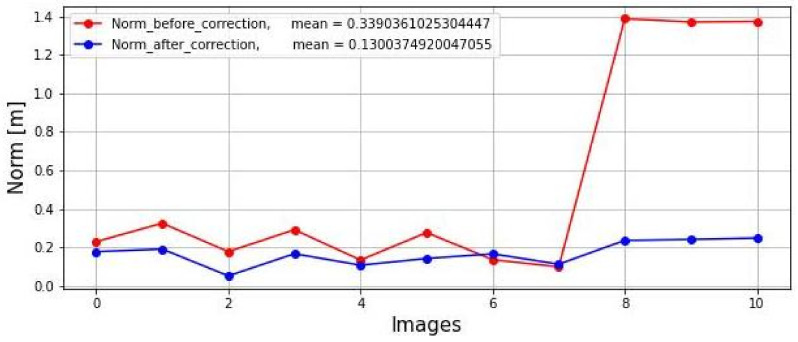
Shows the norm of both lidar and radar datasets before and after correction.

**Figure 15 sensors-23-04167-f015:**
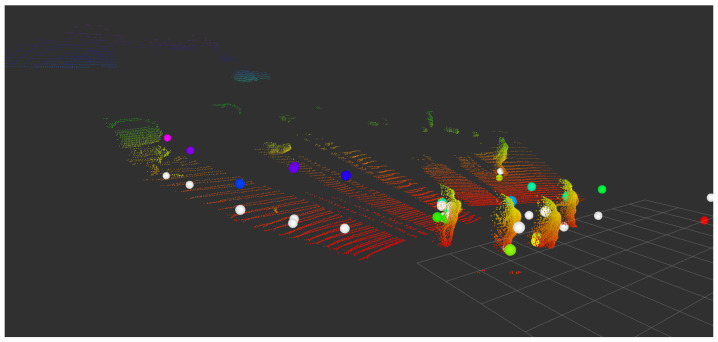
A lidar pointcloud of a parking lot is shown, with white spheres representing the corrected outdoor radar dataset and colored spheres representing the radar dataset before correction.

**Figure 16 sensors-23-04167-f016:**
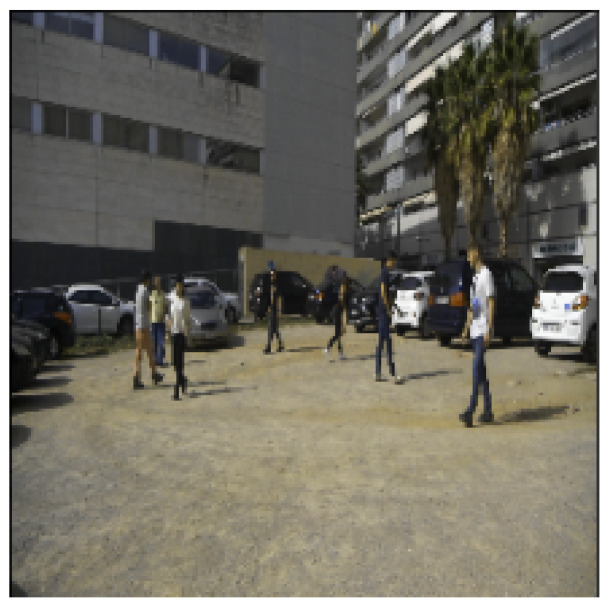
An RGB image of a parking lot shows walking pedestrians.

**Figure 17 sensors-23-04167-f017:**
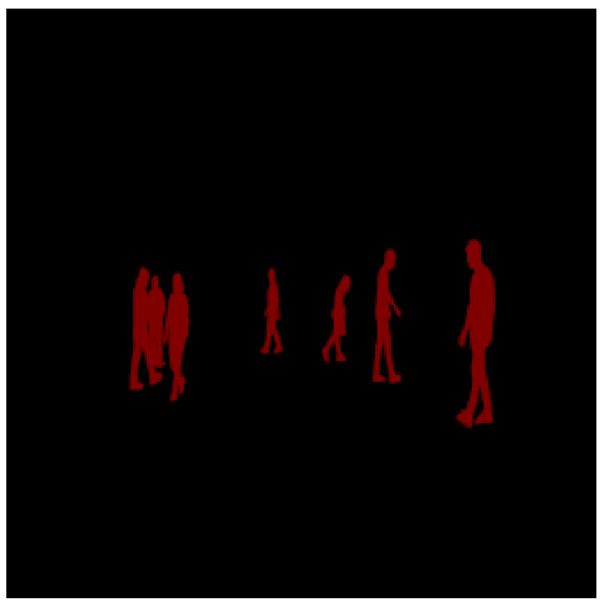
The image represents the ground truth where the pedestrians are shown in red.

**Figure 18 sensors-23-04167-f018:**
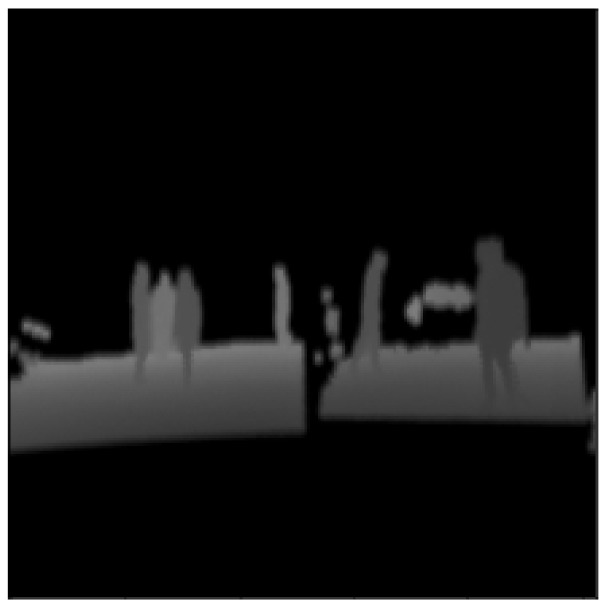
The 3D lidar pointcloud is projected into a 2D grayscale image with 16-bit depth.

**Figure 19 sensors-23-04167-f019:**
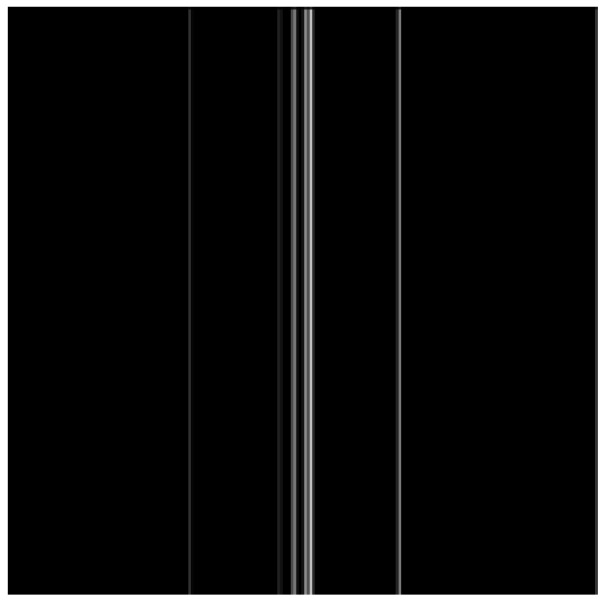
The 3D radar pointcloud is projected into 2D grayscale lines with 16-bit depth.

**Figure 20 sensors-23-04167-f020:**
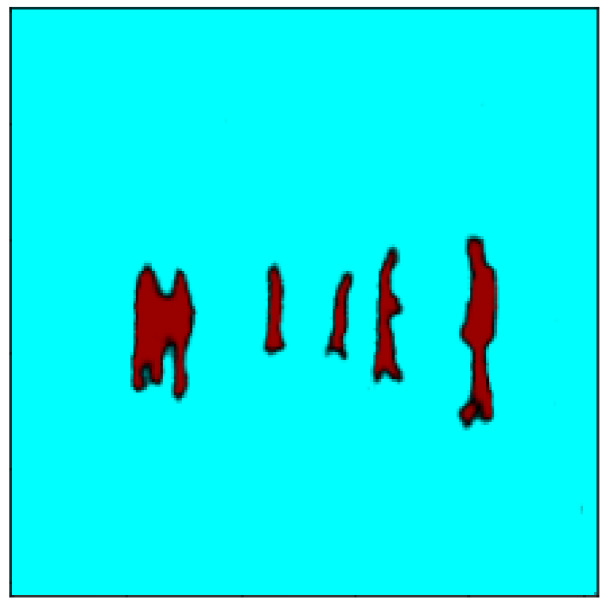
The fusion of the three images corresponding to the lidar, radar, and RGB is shown in red.

**Figure 21 sensors-23-04167-f021:**
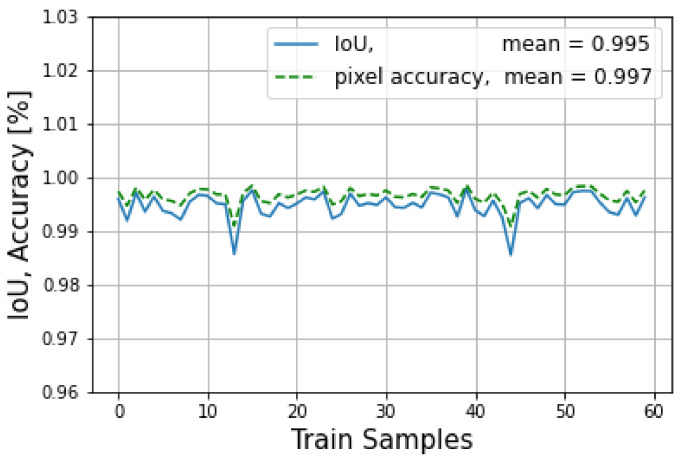
The IoU and pixel accuracy are shown for the training mode.

**Figure 22 sensors-23-04167-f022:**
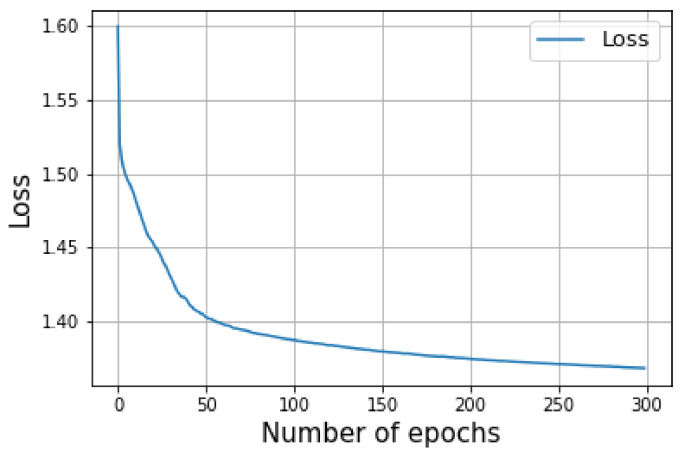
Illustrates the loss entropy.

**Figure 23 sensors-23-04167-f023:**
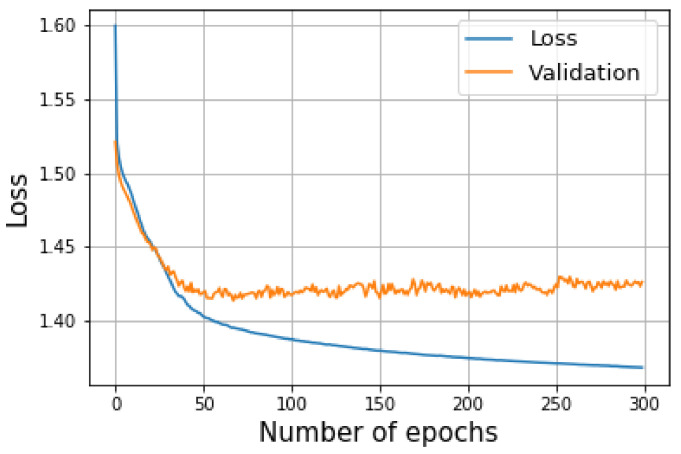
The blue line represents the loss of the training mode and the orange line indicates the loss of the validation mode.

**Figure 24 sensors-23-04167-f024:**
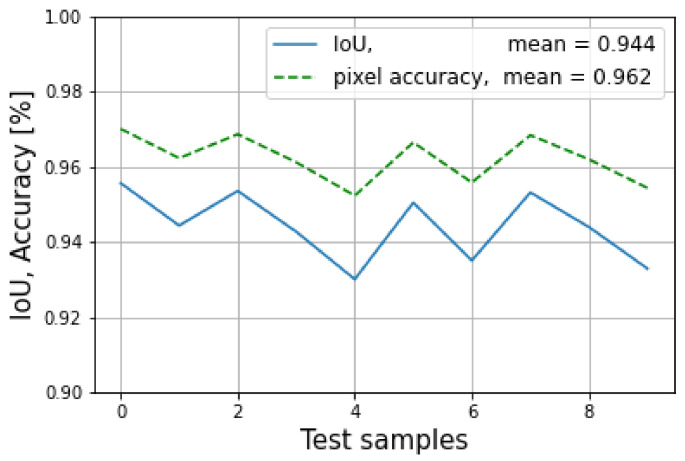
The IoU and pixel accuracy are shown for the testing mode.

**Figure 25 sensors-23-04167-f025:**
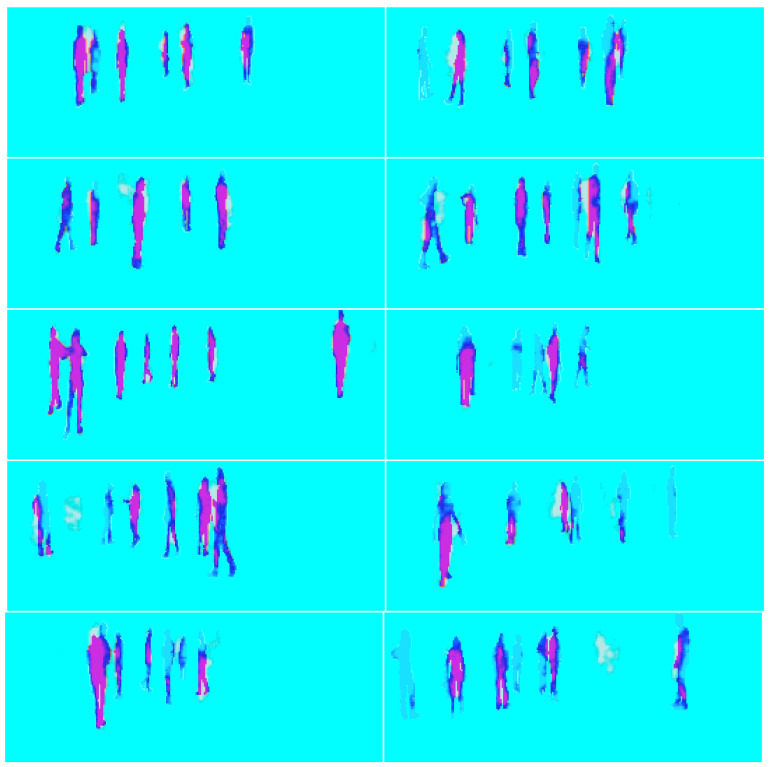
The figure displays in red the overlapping area between the ground truth and the model’s output. The ground truth is depicted in blue, and the white spots are part of the model’s output but do not overlap. The weak blue light over the pedestrians indicates a very weak detection by the model due to its overfitting.

**Table 1 sensors-23-04167-t001:** Encoder network model.

Encoder Network					
Layer	In Channels	Out Channels	Kernel	Padding	Stride
conv 1	3	64	3	1	1
conv 2	64	64	3	1	1
MaxPool		1	2	0	2
conv 3	64	128	3	1	1
conv 4	128	128	3	1	1
MaxPool		2	2	0	2
conv 5	128	256	3	1	1
conv 6	256	256	3	1	1
conv 7	256	256	3	1	1
MaxPool 3			2	0	2
conv 8	256	512	3	1	1
conv 9	512	512	3	1	1
conv 10	512	512	3	1	1
MaxPool 4			2	0	2
conv 11	512	512	3	1	1
conv 12	512	512	3	1	1
conv 13	512	512	3	1	1
MaxPool 5			2	0	2

**Table 2 sensors-23-04167-t002:** Fully connected layer.

Fully Connected Network		
Layer	In	Out
fc1	98,304	2048
fc2	2048	1024
fc3	1024	65,536

**Table 3 sensors-23-04167-t003:** Decoder network model.

Decoder Network					
Layer	In Channels	Out Channels	Kernel	Padding	Stride
MaxUnpool 1			2	0	2
conv 1	512	512	3	1	1
conv 2	512	512	3	1	1
conv 3	512	512	3	1	1
MaxUnpool 2			2	0	2
conv 4	512	512	3	1	1
conv 5	512	512	3	1	1
conv 6	512	256	3	1	1
MaxUnpool 3			2	0	2
conv 7	256	256	3	1	1
conv 8	256	256	3	1	1
conv 9	256	128	3	1	1
MaxUnpool 4			2	0	2
conv 10	128	128	3	1	1
conv 11	128	64	3	1	1
MaxUnpool 5			2	0	2
conv 12	64	64	3	1	1
conv 13	64	3	3	1	1

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
