# Peer review of "A Preliminary Study of Deep Learning Sensor Fusion for Pedestrian Detection"

_sensors, 2023, doi:10.3390/s23084167_

Round 1

Reviewer 1 Report

The manuscript proposes a SegNet-based CNN model for the fusion of three sensor modalities (lidar, sonar, and RGB camera) for pedestrian detection in autonomous driving.

Following are the comments the authors should address in the revised manuscript:

1.      The Abstract is focused solely on the technical aspect of the method proposed in the paper. The authors should provide the research background and the main problems that need addressing at the beginning of the Abstract to explain the motivation for this study. Moreover, the Abstract gives no information on the application of the proposed method (pedestrian detection in autonomous driving). Finally, the Abstract should contain more factual information about the obtained results.

2.      The authors should add a paragraph detailing the manuscript’s organization to the end of the Introduction section.

3.      The authors should emphasize more the contribution of this study in comparison with the reference [14].

4.      The literature review is relatively short. The authors should extend it by addressing some recent review papers on the application of deep learning and lidar data in autonomous driving, including multimodal sensor fusion and pedestrian detection (10.3390/su12083281, 10.3390/s22165946, 10.3390/s20072068).

5.      The dataset of 30 RRL images used for CNN training is rather small. The authors should elaborate more on the dataset size appropriateness. Moreover, the authors should provide and elaborate on arguments that no overfitting of the CNN model occurred.

6.      In the Conclusion section, when commenting on the achieved results, the authors should be more concrete and use values of the obtained quantitative evaluation metrics.

Reviewer 2 Report

The authors developed a deep learning based sensor fusion algorithm to detect the pedestrian. They investigated the aspects of fusing RGB images with lidar and radar pointclouds for pedestrian detection using the novel and practical deep fully connected CNN architecture for semantic pixel-wise segmentation. The authors have discussed it comprehensively by getting some important things presented in some figures and their contributions to the literature are nice. But, there are many things that need fixing. I think this paper needs Major Revision before acceptance process. Necessary changes suggested to be made are listed below.

1. The paper involves a lot of grammar and spelling mistakes. The authors should recheck all of the paper. For example; “3. Proposed Archtecture”, the last word at line 105 etc.

2. For example, the sentence at Line 121 and 122 involves a mistake about citing equations in the text. Please recheck all of the text. 

3. They should use passive sentences. Please recheck the text. 

4. Abstract section was not written well. The author should summarize the background of the problem and their motivation. The authors should explain original scientific contributions and provide some information about the advantages or benefits of the presented methodology. Also, the authors may add one or two sentences about their findings.

5. The introduction was not well-written and the references were not sorted very well. I can’t see the references 7,8,9, etc… The authors missed citing a lot of references in the text. 

6. The authors should increase the number of the references and add more related studies about the sensor fusion techniques for detection of pedestrians and other obstacles on the road. 

7. At the end of the introduction part, the authors should explain clearly the significance and importance of this work by referring to previously published studies.

8. The authors should rewrite the captions of the figures. For example, they were written as “ Figure 1. Shows the RGB camera, lidar and radar system.”. I think these captions should be improved academically. 

9. There are a lot of spaces before and after the tables and figures. Please, adjust these spaces according to the journal format. 

10. The language of the paper should be improved academically. The English level of the paper should be also improved for increasing the readability and understandability. 

11. The font of the equations at Line 106 is different from the journal template. Please check whole text.

12. Conclusion section should be expanded and rewritten. The authors should place a larger emphasis on this section. In this section,  some of the key findings should be briefly explained.

Reviewer 3 Report

The author just introduce their work and I cannot find any new ideas or technologies. The following are some comments:

1.      The performance of detector designed by CNN relies greatly on the dataset. Some recent studies (A Combined Object Detection Method With Application to Pedestrian Detection. IEEE Access, 2020,8:194457-194465) about the fusion of learned and designed features together to improve the detection performance under unknown conditions are suggested to be summarized and discussed in the introduction.

2.      For AV, the real time performance is critical, which should be presented.

3.      The detail information about the dataset used to train the network should be presented.

4.      More details about the fusion of different sensors and the special considerations for AV should be presented.

5.      It is suggested to compare with other latest detection algorithms, such as only vision, methods for lidar and even fusion methods.

Author Response

Reviwer 3:
Comments and Suggestions for Authors
The author just introduce their work and I cannot find any new ideas or technologies. The following
are some comments:
1. The performance of detector designed by CNN relies greatly on the dataset. Some recent studies
(A Combined Object Detection Method With Application to Pedestrian Detection. IEEE Access,
2020,8:194457-194465) about the fusion of learned and designed features together to improve
the detection performance under unknown conditions are suggested to be summarized and
discussed in the introduction.
The previous reference has been added in the introduction, in lines 82-87.
2.For AV, the real time performance is critical, which should be presented.
In line 303, it is stated that,
the mean average of the model in testing mode was 0.010 seconds.
3.The detail information about the dataset used to train the network should be presented.
An explanation of the dataset is presented starting from line 275.
4.More details about the fusion of different sensors and the special considerations for AV should be
presented.
This is a good point taken, the introduction was enhanced where more references are added.
5.It is suggested to compare with other latest detection algorithms, such as only vision, methods for
lidar and even fusion methods.
As mentioned in the previous point, the introduction was enhanced with more references, including
sensor fusion techniques involving CNN, sensor fusion approaches based on semantic
segmentation, detection of pedestrians using classical methods, and object detection for
autonomous vehicles.We appreciate the thorough review you have conducted and the improvements you have
suggested. The authors believe that your proposals have increased the readability of the
article and we have incorporated them, but we want to specifically thank you for your
detailed review.
Regards,
The authors

Reviewer 4 Report

Summary:

I enjoyed reading the article "Deep Learning Sensor Fusion for Pedestrian Detection" by Plascencia et al. The article documents a useful research study, and will, in my opinion, be interesting to the readers of MDPI Sensors after all necessary revisions are made. It's always a pleasure for me to read articles typeset in LaTeX.

The authors propose a method that uses a sensor fusion pixel semantic segmentation network to detect pedestrians by fusing the output of a lidar, a radar and a video camera. The authors describe, both formally and informally, a calibration method to find the external parameters between their lidar and radar models. A custom dataset was collected at a parking lot. The dataset consists of 60 images for training, 10 images for validation and 10 for testing. The authors report that the training mean pixel accuracy is 99.77% and the training mean intersection over union is 99.66%. The testing mean of the intersection over union is reported at 96.22 % and the testing mean pixel accuracy is reported at 97.45%.

--------------

Major Comments:

1) This research is a pilot study, because the image dataset is very small. The proposed network model overfits during validation. While it does detect pedestrians in the testing mode, it needs to be tested on the pixels of images obtained from a different parking lot or a different site which not used in training at all. While the authors state that the focus of their research is to show that the method "works" despite the size of the dataset, it does not change the pilot nature of this research. The current abstract does not use the terms "pilot" or "preliminary." These terms should be added. As
written, the abstract does not reflect the scope of the investigation.

2) The authors state that their method has the advantage of seeing pedestrians the same way as the human eye does. This theme is not developed adequately in the article. Perhaps, it can be dropped altogether. If the authors choose to retain this theme, they should refine it with references from cognitive science and psychovision. Furthermore, since no comparison is done on the same dataset with competitive approaches, there is no factual evidence in the article that the performance of the proposed method is better than any of the alternative approaches (i.e., conventional methods of pedestrian detection with bounding boxes). This is another reason to state
clearly in the abstract of the article and re-iterate in the discussion that this
research is pilot and the results are preliminary and should be interpreted with
caution.

3) The article should be re-structured. Some paragraphs in the introduction
section should be moved into the discussion. See my specific comments below. The conclusion section is missing and must be added. The descriptions
of Algorithms 1 and 2 must be improved.

--------------

Specific Comments:

1) Line 27: Remove the word "ability" in front of "scalability. It's redundant.

2) Line 33: "the leading towards a fully customized public car is getting
closer than ever."

This is not a grammatically correct sentence. Did you mean to say that
"fully customized public cars will likely become a reality soon." Did you
mean to write "fully automated" instead of "fully customized"?

3) Line 53: "can not" -> "cannot"

4) Line 54: "which" -> "whose" or "where"

5) Line 61: "literature" -> "the literature"

6) Line 70: "CNN's" -> "CNNs"; "CNN's" is possessive. I don't think you
meant as a possessive case in this sentence.

7) Lines 76-77: Move this sentence to the discussion. It is a critique
of an alternative approach. In general, any critique of the alternative
approaches should occur in the discussion. The purpose of the introduction
section is to introduce the reader into the problem and its background
by giving references to the state of the art.

8) Line 94: "However, human eyes detect objects with pixel-level precision
at the boundary level."

Do you have citations to back up this statement, or is this your hypothesis?
If the latter is the case, then state it clearly. If the former, then you
should provide a citation. On line 90 you cite [20, 21]. If these publications
make or validation this hypothesis, then you can move one of them to line 94.

9) Lines 99-100: remove comma after "where," and  change "ach" to "each."

10) Line 102: "The stream data is then fused by a fully connected layer,
which outputs are up sampled by a decoder layer that recovers the fused data."

Could you briefly explain why the fused data should be recovered? Is this
done for testing purposes by aligning classified pixels with the ground truth
in Figure 25 on p. 15?

11) Line 115-16: "As mentioned earlier, most 3D methods for representing
objects and people rely on bounding boxes, which is not the best
representation [23]."

You should move this sentence to the discussion. The introduction
section should contain only the factual background information. The
critique of peer work belongs in the discussion.

12) Line 120: "The article is divided as follows:" should be changed to
"The remainder of this article is organized as follows." Note the period
at the end.

13) The reference to the github repo should be either a reference or
provided in the supplementary materials.

14) At the end of the introduction section you should summarize, in
a paragraph, the main contributions of your work.

15) Caption of Figure 1 on p. 4. "b) At the bottom, the UMRR-96 Type 153
radar sensor."

Did you mean "to the right" instead of "at the bottom"?

16) Lines 137-139.

You may want to get rid of these lines. They don't contribute any
content to the article. The structure of this section is clear
as is.

17) Line 174: 2.2.1. First Step (image point intersection)

Change to "Step 1: Image Point Segmentation"

18) Line 180: "Where," --> "where"

19) 2.2.2. Second Step (lidar point)

Change to "Step 2: Lidar Point"

20) Line 184: "The Algorithm 1"

Change it to either "The first algorithm" or "Algorithm 1" w/o the definite
article.

21) Line 185: "the point (x,y) is undistorted..."

Could you briefly explain how the undistortion of points is computed?

22) Line 192: "The function PlaneLineCrossingPoint(.) ..."

What is this function? Where is it defined? Is it part of the sensor
software?

23) Line 193: "2.2.3. Third Step (extrinsic calibration matrix)"

Change to "2.2.3. Step 3: Extrinsic Calibration Matrix"

24) What is [1] in the description of Algorithm 1 on p. 7?

25) The description of Algorithm 1 on p. 7 was really difficult to read
for me. Perhaps, you can add \begin{enumerate} \item ... \end{enumerate}
to clearly identify each step. You can describe the algorithm as a sequence
of function calls and then explain in the subsequent or preceding text
what each function does.

26) Line 203: Remove "The" before "Algorithm 2."

27) The description of Algorithm 2 on p. 7 was really difficult to read
for me. Perhaps, you can add \begin{enumerate} \item ... \end{enumerate}
to clearly identify each step. You can describe the algorithm as a sequence
of function calls and then explain in the text what each function does.
Why do you even need an algorithm box here?. You can replace the algorithm
boxes with a series of equations and then explain what each equation does
in the text.

28) Figures 7 and 8 are really nice. In general, all Figures are well done.

29) Lines 266-269:  "The dataset was taken in the format
yyyy-MM-dd:hh:mm:ss:zz, where yyyy denotes the year, MM the month, dd
the day, hh the hour, mm the minute, ss the second, and zz the
milliseconds. This format was chosen because the lidar, RGB, and radar
frequencies are 10 Hz, 6 Hz, and 18 Hz, respectively."

What does the time format have to do with the frequencies? Could you clarify?

30) Lines 277-280: "Training, evaluating and testing modes were executed on Jupyter-Notebook with Pytorch version 1.11 and GPU activated. The RRL images were downsampled to a size of 256 x 256. Moreover, due to the capacity of the GPU which is 8 GB, a batch size was set to one. Next, the network was trained 300 epochs."

How long did the training take in terms of physical time? Hours? Days? Weeks? Please state the amount of physical time.

31) Lines 280-281: Additionally, 10 RRL images were used for evaluation and 10 RRL for testing, giving a total dataset of 80 RRL.

Did you mean to say "validation" instead of "evaluation"? A typical neural network dataset consists of training, validation and testing sub-datasets.

32) Figures 18, 19 on p. 13. "grays cale" -> "grayscale."

33) Line 286: To evaluate the results of the Network in training mode,

What does it mean? Why would you want to evaluate the network while it's training? Is this to show that the loss is higher during testing? But, that's almost always the case with convolutional networks.

34) The notation in Equations (10) and (11) can be improved. Put the semicolon after (10) and a comma after (11), then change Line 291 to read

"where the true positive (TP) is the area ..."

35) I'm not sure what "if" is in Equation (12). Please specify.

36) Line 298: "Loss" -> "loss"

37) I became really confused when I was looking at Figure 21 on p. 14. What exactly is the neural network being trained on? Isn't it trained on pixels, not images? Earlier in the article you write that "each pixel is classified individually and assigned to a class that best represents it."  In other words, if an image is 256x256, then the network is trained on 256^2 = 65,536 pixels per image, correct? Equations (10) and (11) are defined in terms of pixels. Please clarify.

38) Figure 23 on p. 14: Did you mean "validation" instead of "evaluation"?

39) The discussion section should be re-written. As is, it reads like a summary that essentially repeats the abstract. This is where you should compare your work with the state of the art in the literature and the competitive approaches and discuss their relative advantages and disadvantages.

40) You should write a conclusions section. What did you learn from your work? What are the major takeaways for your fellow researchers and practitioners?

41) The references need to be formatted. But, this will probably be done by an MDPI production editor after the article is ready for publication. I haven't seen
this format: Beamagine. @ONLINE.

Author Response

Reviwer 4:
Major Comments:
1) This research is a pilot study, because the image dataset is very small. The proposed network
model overfits during validation. While it does detect pedestrians in the testing mode, it needs to be
tested on the pixels of images obtained from a different parking lot or a different site which not used
in training at all. While the authors state that the focus of their research is to show that the method
"works" despite the size of the dataset, it does not change the pilot nature of this research. The
current abstract does not use the terms "pilot" or "preliminary." These terms should be added. As
written, the abstract does not reflect the scope of the investigation.
Good point taken, thank you.
The method was tested with a different sample data than the one used in the training mode as it can
be seen in the result section.
We have changed the objective of the paper to a preliminary step study method. We started by
changing the title to: "A preliminary Study of Deep Learning Sensor Fusion for Pedestrian
Detection." Also, we emphasized in the abstract, introduction, and conclusion that it is a preliminary
approach.
2) The authors state that their method has the advantage of seeing pedestrians the same way as the
human eye does. This theme is not developed adequately in the article. Perhaps, it can be dropped
altogether. If the authors choose to retain this theme, they should refine it with references from
cognitive science and psychovision. Furthermore, since no comparison is done on the same dataset
with competitive approaches, there is no factual evidence in the article that the performance of the
proposed method is better than any of the alternative approaches (i.e., conventional methods of
pedestrian detection with bounding boxes). This is another reason to state
clearly in the abstract of the article and re-iterate in the discussion that this
research is pilot and the results are preliminary and should be interpreted with
caution.
In line 89-90 the text was changed.
It is meant that humans requires to detect pedestrians at the pixel level to understand better their
surroundings, according to [20].
We added in lines 4-6
Therefore, the motivation of this work is to explore as a preliminary step the feasibility of fusing
lidar, radar, and RGB for pedestrian detection that potentially can be used for autonomous driving
using a fully convolutional neural network architecture for multimodal sensors.
We also added in the conclusions, in line 340-343.
Experimental results using a custom dataset showed that the proposed architecture
achieved high accuracy and performance, making it suitable for systems with limited
computational resources. But, since this is a preliminary study, the results must be
interpreted with caution.
3) The article should be re-structured. Some paragraphs in the introduction
section should be moved into the discussion. See my specific comments below. The conclusionsection is missing and must be added. The descriptions
of Algorithms 1 and 2 must be improved.
The conclusion section has been added, and algorithms 1 and 2 are as requested. We believe that the
sensor version has altered the structure of the algorithms, as demonstrated in the figures presented
in the following text.
--------------
Specific Comments:
1) Line 27: Remove the word "ability" in front of "scalability. It's redundant.
The word has been removed
2) Line 33: "the leading towards a fully customized public car is getting
closer than ever."
This is not a grammatically correct sentence. Did you mean to say that
"fully customized public cars will likely become a reality soon." Did you
mean to write "fully automated" instead of "fully customized"?
thank you, indeed, it was meant to be “fully automated”
so the sentence was change to:
the leading towards fully automated public cars will likely become a reality soon
3) Line 53: "can not" -> "cannot"
It has changed to cannot
4) Line 54: "which" -> "whose" or "where"
We could not find of the previous word in line 54.
5) Line 61: "literature" -> "the literature"
It has been changed to the literature
6) Line 70: "CNN's" -> "CNNs"; "CNN's" is possessive. I don't think you
meant as a possessive case in this sentence.
Thanks you for the correction, it was meant to be plural, like CNNs. So, it has been changed to
CNNs
7) Lines 76-77: Move this sentence to the discussion. It is a critique
of an alternative approach. In general, any critique of the alternative
approaches should occur in the discussion. The purpose of the introduction
section is to introduce the reader into the problem and its background
by giving references to the state of the art.
Well, we consider to delete it instead of moving it to the discussion section.
However, the networks were trained using different datasets and tools, so the comparison may not
be very accurate.8) Line 94: "However, human eyes detect objects with pixel-level precision
at the boundary level."
Do you have citations to back up this statement, or is this your hypothesis?
If the latter is the case, then state it clearly. If the former, then you
should provide a citation. On line 90 you cite [20, 21]. If these publications
make or validation this hypothesis, then you can move one of them to line 94.
It was meant to say according to the reference [21] that the more the autonomy of cars the better to
detect humans at the pixel level, so the sentence was chanced to:
However, humans requires to detect humans at the pixel level to understand better their
surroundings, specially in the process of building autonomous cars, [21].
9) Lines 99-100: remove comma after "where," and change "ach" to "each."
It is done:
where each sub-network is assigned to a particular sensor stream
10) Line 102: "The stream data is then fused by a fully connected layer,
which outputs are up sampled by a decoder layer that recovers the fused data."
Could you briefly explain why the fused data should be recovered? Is this
done for testing purposes by aligning classified pixels with the ground truth
in Figure 25 on p. 15?
In the line 108 a sentence that explains the upsample is done according to reference [25] is added.
The upsampling is done in order to recover the lost spatial information and generate a dense
pixel-wise segmentation map that accurately captures the fine-grained details of the input
image, [25].
11) Line 115-16: "As mentioned earlier, most 3D methods for representing
objects and people rely on bounding boxes, which is not the best
representation [23]."
You should move this sentence to the discussion. The introduction
section should contain only the factual background information. The
critique of peer work belongs in the discussion.
This reasoning is moved to the conclusion section.
12) Line 120: "The article is divided as follows:" should be changed to
"The remainder of this article is organized as follows." Note the period
at the end.
Thank you, we have changed that sentence.
13) The reference to the github repo should be either a reference or
provided in the supplementary materials.We have moved the github to the reference section.
14) At the end of the introduction section you should summarize, in
a paragraph, the main contributions of your work.
Thank you, a paragraph is added at the end of the introduction section:
Thus, the main contribution of this paper is to explore the feasibility of fusing RGB
images with lidar and radar pointclouds for pedestrian detection using a small dataset. It
is widely recognized that larger datasets lead to better training, but small datasets can
result in overfitting [26]. Nevertheless, studies [27,28] have shown that data size is not an
obstacle to high-performing models. To address the previous, a novel and practical deep
fully connected CNN architecture for semantic pixel-wise segmentation called SegNet [25]
is proposed. The network consists of three SegNet sub-networks that down sample the
inputs of each sensor, a fully connected (fc) neural network (NN) that fuses the sensor data,
and a decoder network that up samples the data. Therefore, the proposed method focuses
on detecting people at a pixel level. The task of identifying pedestrians is referred to as
semantic segmentation and involves producing pixel-level classifications based on a dataset
that has been labeled at the pixel level. Typically, there is only one class of interest, namely
pedestrians. Moreover, the inclusion of radar in the fusion process gives the advantage of
being able to detect pedestrians in severe weather conditions. Additionally, an extrinsic
calibration method for radar with lidar and RGB camera, based on the work done in [29,30],
is presented.
15) Caption of Figure 1 on p. 4. "b) At the bottom, the UMRR-96 Type 153
radar sensor."
Did you mean "to the right" instead of "at the bottom"?
Thank you, it means to the bottom, because there are two sensors, the one in the top which is the
laser + the camera and the one in the bottom which is the radar.
16) Lines 137-139.
You may want to get rid of these lines. They don't contribute any
content to the article. The structure of this section is clear
as is.
It si right, those lines have been removed.
17) Line 174: 2.2.1. First Step (image point intersection)
Change to "Step 1: Image Point Segmentation"
It has been changed.
18) Line 180: "Where," --> "where"
Done
19) 2.2.2. Second Step (lidar point)Change to "Step 2: Lidar Point"
It has been changed.
20) Line 184: "The Algorithm 1"
Change it to either "The first algorithm" or "Algorithm 1" w/o the definite
article.
It has been changed to “The first algorithm”
21) Line 185: "the point (x,y) is undistorted..."
Could you briefly explain how the undistortion of points is computed?
A small paragraph has been added at line 193.
First, the OpenCV undistortPoints function is used to remove distortion from a set of image
points ( x, y ) . The function takes distorted points, camera matrix, and distortion coefficient
as input and outputs undistorted points ( u , v ) .
22) Line 192: "The function PlaneLineCrossingPoint(.) ..."
What is this function? Where is it defined? Is it part of the sensor
software?
Well, the function PlaneLineCrossingPoint() is part of the pseudo code of the first algorithm. It is
not part of the sensor software.
23) Line 193: "2.2.3. Third Step (extrinsic calibration matrix)"
Change to "2.2.3. Step 3: Extrinsic Calibration Matrix"
It is done
24) What is [1] in the description of Algorithm 1 on p. 7?
25) The description of Algorithm 1 on p. 7 was really difficult to read
for me. Perhaps, you can add \begin{enumerate} \item ... \end{enumerate}
to clearly identify each step. You can describe the algorithm as a sequence
of function calls and then explain in the subsequent or preceding text
what each function does.
The algorithm is organized as you suggested, I think the version from the sensors somehow modify
the structure. It can be seen in the next Figure.26) Line 203: Remove "The" before "Algorithm 2."
It has been removed.
27) The description of Algorithm 2 on p. 7 was really difficult to read
for me. Perhaps, you can add \
begin{enumerate} \item ... \end{enumerate}
to clearly identify each step. You can describe the algorithm as a sequence
of function calls and then explain in the text what each function does.
Why do you even need an algorithm box here?. You can replace the algorithm
boxes with a series of equations and then explain what each equation does
in the text.
I think, this is the same as the algorithm 1, it is organized as suggested, as it can be seen in the
following Figure.28) Figures 7 and 8 are really nice. In general, all Figures are well done.
29) Lines 266-269: "The dataset was taken in the format
yyyy-MM-dd:hh:mm:ss:zz, where yyyy denotes the year, MM the month, dd
the day, hh the hour, mm the minute, ss the second, and zz the
milliseconds. This format was chosen because the lidar, RGB, and radar
frequencies are 10 Hz, 6 Hz, and 18 Hz, respectively."
What does the time format have to do with the frequencies? Could you clarify?
The following paragraph was added at the line 275.
The CNN-Network was trained using a custom dataset that was taken at a parking lot with people
walking in front of the system. The dataset was taken in the format yyyy-MM-dd:hh:mm:ss:zz,
where yyyy denotes the year, MM the month, dd the day, hh the hour, mm the minute, ss the second,
and zz the milliseconds. This format was chosen because the lidar, RGB, and radar frequencies are
6 Hz, 10 Hz, and 18 Hz, respectively. For instance, a typical format is 20221017_131347_782.pcd
where hh=13, mm=13, and ss=48. This means 48 will be repeated 6 times for the lidar, 10 times for
the RGB, and 18 times for the radar. The value of zz=782 will be different for each reading, making
it easier to synchronize them.
30) Lines 277-280: "Training, evaluating and testing modes were executed on Jupyter-Notebook
with Pytorch version 1.11 and GPU activated. The RRL images were down sampled to a size of 256
x 256. Moreover, due to the capacity of the GPU which is 8 GB, a batch size was set to one. Next,
the network was trained 300 epochs."
How long did the training take in terms of physical time? Hours? Days? Weeks? Please state the
amount of physical time.In line 302, it is stated that, the time elapsed during the training of the network was 7019.40
seconds.
31) Lines 280-281: Additionally, 10 RRL images were used for evaluation and 10 RRL for testing,
giving a total dataset of 80 RRL.
Did you mean to say "validation" instead of "evaluation"? A typical neural network dataset consists
of training, validation and testing sub-datasets.
Thank you, validation was meant, it has been changed.
32) Figures 18, 19 on p. 13. "grays cale" -> "grayscale."
It is done.
33) Line 286: To evaluate the results of the Network in training mode,
What does it mean? Why would you want to evaluate the network while it's training? Is this to show
that the loss is higher during testing? But, that's almost always the case with convolutional
networks.
The previous is explained from the line 303.
The target and prediction during training are stored in a vector. Later on, the
intersection over union (IoU) is evaluated for each frame to see the percentage of the
overlapping area between the prediction and the target. To this end, to evaluate the results
of the network in training mode, pixel accuracy and intersection over union (IoU) metrics
for image segmentation are used. Additionally, the cross-entropy loss function is applied.
34) The notation in Equations (10) and (11) can be improved. Put the semicolon after (10) and a
comma after (11), then change Line 291 to read
"where the true positive (TP) is the area …"
It is done.
35) I'm not sure what "if" is in Equation (12). Please specify.
From the line 316 the following was added.
In this equation if the confusion vector is equal to 1 means that there is an overlapping, then the
TP is computed by adding all ones, the same happen for FP, TN and FN.
36) Line 298: "Loss" -> "loss"
It is done.
37) I became really confused when I was looking at Figure 21 on p. 14. What exactly is the neural
network being trained on? Isn't it trained on pixels, not images? Earlier in the article you write that
"each pixel is classified individually and assigned to a class that best represents it." In other words,
if an image is 256x256, then the network is trained on 256^2 = 65,536 pixels per image, correct?
Equations (10) and (11) are defined in terms of pixels. Please clarify.A paragraph in the lines 124-128 has been added.
Therefore, the proposed method focuses on detecting people at a pixel level. The task of identifying
pedestrians is referred to as semantic segmentation and involves producing pixel-level
classifications based on a dataset that has been labeled at the pixel level. Typically, there is only one
class of interest, namely pedestrians.
Also a sentence is added in line 312.
Both Equations 10 and 11 are defined in terms of pixel accuracy.
38) Figure 23 on p. 14: Did you mean "validation" instead of "evaluation"?
Validation was meant, thank you.
39) The discussion section should be re-written. As is, it reads like a summary that essentially
repeats the abstract. This is where you should compare your work with the state of the art in the
literature and the competitive approaches and discuss their relative advantages and disadvantages.
40) You should write a conclusions section. What did you learn from your work? What are the
major takeaways for your fellow researchers and practitioners?
It is consider just to add a conclusion section, from lines 334-351. The conclusion section was
rewritten.
41) The references need to be formatted. But, this will probably be done by an MDPI production
editor after the article is ready for publication. I haven't seen
this format: Beamagine. @ONLINE.
We appreciate the thorough review you have conducted and the improvements you have
suggested. The authors believe that your proposals have increased the readability of the
article and we have incorporated them, but we want to specifically thank you for your
detailed review.
Regards,
The authors

Reviewer 5 Report

This manuscript proposed a convolutional neural network architecture for pedestrian detection using sensor fusion applications including LIDAR, RADAR, and camera. In particular, the authors applied the model to real-world experiments. Detailed sensor calibration steps are well described. Overall, the manuscript is well written and only needs some minor change. Before proceeding to publication, I encourage the authors to address my following comments / questions:

1.      [Abstract in system] Is SONAR a typo for RADAR?

2.      [Figure 21-24] Please label x and y axis with proper units.

3.      How did you generate ground truth Figure 17 and Figure 25?

4.      Have authors also conducted experiments in the cases where the sensor system location is not stationary? Does the same calibration process still apply?

Author Response

Reviwer 5:
Comments and Suggestions for Authors
This manuscript proposed a convolutional neural network architecture for pedestrian detection
using sensor fusion applications including LIDAR, RADAR, and camera. In particular, the authors
applied the model to real-world experiments. Detailed sensor calibration steps are well described.
Overall, the manuscript is well written and only needs some minor change. Before proceeding to
publication, I encourage the authors to address my following comments / questions:
1. [Abstract in system] Is SONAR a typo for RADAR?
Well, Both sensors are useful for detecting distances to objects, the radar mainly works with
electromagnetic waves whereas the sonar works with sound waves. However, since the abstract has
been modified, the word of sonar is not longer in the abstract.
2.[Figure 21-24] Please label x and y axis with proper units.
It is done.
3. How did you generate ground truth Figure 17 and Figure 25?
The following paragraph was added in lines 286-288
The program ’labelme’ was used to label the RGB images as ’single-class person.’ In other words,
’labelme’ can create polygons of the persons and save them as a JSON, which can be extracted as a
PNG image with a single person class.
4.Have authors also conducted experiments in the cases where the sensor system location is not
stationary? Does the same calibration process still apply?
During the gathering of the data, the sensor was stand still and the pedestrians where walking
randomly in front of it. Concerning about the moving of the sensor’s system, it is considered in a
further work as it is stated in the conclusion section, line 328.
For future work, investigating the behavior of the proposed architecture with a larger dataset
and testing the model in different weather conditions is needed. Additionally, evaluating the
architecture under a real scenario by mounting the sensors on a driving car shall beinvestigated. Overall, this preliminary study provides a promising starting point for further
research and development in the field of sensor fusion for pedestrian detection.
We appreciate the thorough review you have conducted and the improvements you have
suggested. The authors believe that your proposals have increased the readability of the
article and we have incorporated them, but we want to specifically thank you for your
detailed review.
Regards,
The authors

Round 2

Reviewer 1 Report

Referring to the authors' response, the presented study is not technically sound, i.e. the utilized dataset is too small to enable appropriate training and to allow drawing valid conclusions about the method's performance without serious danger of overfitting effects.

Moreover, the manuscript's novelty and contributions to the existing research are rather limited.

Author Response

Reviwer 1:
We have changed the objective of the paper to a preliminary study method. We started by changing
the title to: "A Preliminary Study of Deep Learning Sensor Fusion for Pedestrian Detection."
Also, we emphasized in the abstract, introduction, and conclusion that it is a preliminary approach.
Although we acknowledge that the data set is small and may result in over-fitting, larger data sets have
the potential to improve training. Nonetheless, as demonstrated in references [25,26], data size is not a
hindrance to high-performing models. By drawing from previous research, we present our method as a
pilot study that can be improved upon with a larger data set. Furthermore, we have demonstrated the
avoidance of overfitting and used the model at its optimal performance, making it suitable for systems
with limited computational resources.
We appreciate the thorough review you have conducted and the improvements you have
suggested. The authors believe that your proposals have increased the readability of the
article and we have incorporated them, but we want to specifically thank you for your
detailed review.
Regards,
The authors

Reviewer 2 Report

I have no more comment on the paper. It can be accepted.

Author Response

Reviwer 2:
We appreciate the thorough review you have conducted and the improvements you have
suggested. The authors believe that your proposals have increased the readability of the
article and we have incorporated them, but we want to specifically thank you for your
detailed review.
Regards,
The authors

Reviewer 3 Report

The author has addressed all my concerns.